# Dysregulated Cardiac IGF-1 Signaling and Antioxidant Response Are Associated with Radiation Sensitivity

**DOI:** 10.3390/ijms21145049

**Published:** 2020-07-17

**Authors:** Saeed Y. Aghdam, Doreswamy Kenchegowda, Neel K. Sharma, Gregory P. Holmes-Hampton, Betre Legesse, Maria Moroni, Sanchita P. Ghosh

**Affiliations:** Armed Forces Radiobiology Research Institute, Uniformed Services University of the Health Sciences, Bethesda, MD 20889, USA; doreswamy.kenchegowda.ctr@usuhs.edu (D.K.); neel.sharma.ctr@usuhs.edu (N.K.S.); gregory.holmes-hampton.ctr@usuhs.edu (G.P.H.-H.); betre.legesse.ctr@usuhs.edu (B.L.); maria.moroni@usuhs.edu (M.M.)

**Keywords:** radiation, IGF-1, cardiovascular, Nrf2, Redox status

## Abstract

Acute exposure to ionizing radiation leads to Hematopoietic Acute Radiation Syndrome (H-ARS). To understand the inter-strain cellular and molecular mechanisms of radiation sensitivity, adult males of two strains of minipig, one with higher radiosensitivity, the Gottingen minipig (GMP), and another strain with comparatively lower radiosensitivity, the Sinclair minipig (SMP), were exposed to total body irradiation (TBI). Since Insulin-like Growth Factor-1 (IGF-1) signaling is associated with radiation sensitivity and regulation of cardiovascular homeostasis, we investigated the link between dysregulation of cardiac IGF-1 signaling and radiosensitivity. The adult male GMP; *n* = 48, and SMP; *n* = 24, were irradiated using gamma photons at 1.7–2.3 Gy doses. The animals that survived to day 45 after irradiation were euthanized and termed the survivors. Those animals that were euthanized prior to day 45 post-irradiation due to severe illness or health deterioration were termed the decedents. Cardiac tissue analysis of unirradiated and irradiated animals showed that inter-strain radiosensitivity and survival outcomes in H-ARS are associated with activation status of the cardiac IGF-1 signaling and nuclear factor erythroid 2-related factor 2 (Nrf2)-mediated induction of antioxidant gene expression. Our data link H-ARS with dysregulation of cardiac IGF-1 signaling, and highlight the role of oxidative stress and cardiac antioxidant response in radiation sensitivity.

## 1. Introduction

Differential inter-strain radiosensitivity has posed as an unresolved enigma in radiobiology and remains vastly unexplored. Understanding the signaling pathways affected by radiation exposure in different cell and tissue types is of fundamental importance for the development of suitable radioprotective mitigating agents and countermeasures. Insulin-like Growth Factor-1 (IGF-1) is considered as the major effector molecule downstream of the pituitary-originated growth hormone (GH) and regulates myriad anabolic and mitogenic functions in many cells and tissues [1]. The IGF-1 polypeptide consists of 70 amino acids and approximately 75% of the total serum IGF-1 is synthesized and secreted by liver in response to GH stimulation. IGF-1 exerts pleiotropic effects in autocrine, paracrine, and endocrine fashions [2] by binding to its cognate IGF-1 receptors on target cells. IGF-1 signal transduction activates several downstream proteins involved in cell survival, proliferation, propagation [3], and confers radioresistance properties to tumor cells [4]. Other researchers have reported an association between aging and an increase in radiation sensitivity [5]. IGF-1 levels in the circulatory system of elderly humans tend to decline compared to juveniles or healthy adults [6]. Given the radioprotective effects of IGF-1 signaling in tumor cells, the reduced IGF-1 levels in elderly individuals could be linked to susceptibility to irradiation injury. The central role of IGF-1 signaling in physical growth, cell survival, cellular proliferation, and differentiation along with its radioprotective effects raises the question whether IGF-1 signaling is involved in the sensitivity to total body irradiation. We have previously showed that Hematopoietic Acute Radiation Syndrome (H-ARS) impacts systemic IGF-1 levels and the cardiac IGF-1 signaling in fatally irradiated minipigs [7]. In the same study, using two different strains of minipig, we showed that radiation sensitivity was inversely correlated with strain-dependent average body size, meaning that minipigs belonging to the strain with natively larger body size were less sensitive to irradiation. The pathophysiological effects of radiation in the heart is evidently triggered by the disruption of redox status and mitochondrial function of the cardiac tissue [8,9,10]. The current data combined with our previous findings using the minipig model of radiation injury show the potential role of IGF-1 signaling in radiation sensitivity caused by cardiovascular ailment.

Perhaps two of the most important modulatory effects of IGF-1 signaling in cardiovascular homeostasis could be regulating the activity of the critical enzyme, endothelial nitric oxide synthase (eNOS), and mitigation of oxidative stress effects in endothelial cells [11]. IGF-1 signaling induces eNOS activation via phosphorylation at the serine-1177 residue leading to the synthesis and release of nitric oxide (NO) by endothelial cells [12,13]. NO, the gaseous vasodilator molecule, with essential roles in vascular homeostasis, mediates vascular tone and blood pressure. IGF-1 signaling activates eNOS indirectly through the phosphatidylinositol 3-kinase (PI3K)/Akt pathway and incubation of cultured endothelial cells with PI3K inhibitors Wortmannin and LY2940002 reduced NO release by approximately 50% [13]. IGF-1 signaling may also mitigate radiation-induced oxidative stress as its deficiency is associated with disruption of nuclear factor erythroid 2–related factor 2 (Nrf2)-mediated antioxidant responses in the vasculature [11,14].

In our previous studies, kinetics of complete blood count (CBC) profiles of H-ARS survivor and decedent animals from both Gottingen minipig (GMP) and Sinclair minipig (SMP) strains were similar [7,15] (unpublished). Thus, disorders caused by hematopoietic effects of radiation exposure do not appear to be the cause of differential radiosensitivity in GMP and SMP. Considering the radioprotective and growth-stimulatory effects of IGF-1 signaling, we speculated that IGF-1 signaling could be responsible for radiation sensitivity. Thus, using biosamples from GMP and SMP strains, we tested whether a difference in body weight and size, presumably resulting from differential inter-strain IGF-1 signaling, could account for sensitivity to irradiation. This model was appropriate for our study since IGF-1 signaling is involved in regulation of physical growth and body size and importantly, the GMP as the comparatively smaller strain is more sensitive to radiation injury than SMP. The SMP are larger in size and on average have body weights that are 33% higher than age matched GMP [7]. We also reported previously that in age-matched animals, the GMP have slightly lower serum levels of IGF-1 than SMP. Moreover, GMP are more vulnerable to metabolic disorders than other strains of minipig and are routinely used as a model to study metabolic syndrome and obesity [16]. The lower serum IGF-1 levels and vulnerability to metabolic syndrome in GMP potentially reflects the disturbance of insulin and IGF-1 signaling.

Heart IGF-1 signaling plays an integral role in the homeostasis and normal physiology of its cellular components, including cardiomyocytes and vascular endothelial cells. IGF-1 signaling also regulates cardiac metabolism, contractility, hypertrophy, autophagy, senescence, and apoptosis [17,18]. Endothelial IGF-1 signaling regulates vital physiological functions such as migration, endothelial progenitor cell proliferation, and tube formation [17,18]. Scientific evidence shows that cardiac IGF-1 deficiency may drive the development of cardiovascular disease via impairing the activities of canonical and non-canonical signaling pathways in the heart [19,20,21,22]. As mentioned above, IGF-1 signaling positively contributes to the maintenance of endothelial physiological properties via regulating the activity of eNOS. Therefore, in our model we tested (1): whether there were any variations in the activation status of the cardiac canonical IGF-1/(PI3K)/Akt pathway between GMP and SMP strains and (2): if irradiation could differentially impact the activation of this pathway in the hearts of both strains.

## 2. Results

### 2.1. Cardiac IGF-1 Signaling Is Differentially Impacted in GMP and SMP by Post-Irradiation Survival Status

To analyze the status of systemic IGF-1 signaling in irradiated survivor and decedent SMPs, we first measured plasma IGF-1 levels in these animals before and after radiation. There was a trend towards increased IGF-1 levels in decedent SMPs compared with survivors at the time of euthanasia (Figure 1). The change in systemic IGF-1 levels in decedent SMP was similar to the irradiated decedent GMP, which was reported previously [7]. Next, we used western blotting to assess the activity of IGF-1 signaling in postmortem heart samples of non-irradiated control, irradiated survivor, and irradiated decedent GMPs and SMPs. IGF-1 receptor activation was evaluated using two different antibodies. One antibody detects the phosphorylated form of the IGF-1 receptor (IGF-1R) at Tyrosine residues 1135/1136 [23] and the other antibody detects total IGF-1R. IGF-1R activation was evaluated by calculating the ratio of the phosphorylated IGF-1R to total IGF-1R. Side-by-side analysis of heart lysates from GMP and SMP showed that compared with non-irradiated SMP, the non-irradiated GMP had higher levels of IGF-1R activation (Figure 2A,B). In GMP survivors, the IGF-1R phosphorylation increased significantly compared to non-irradiated GMP, whereas in decedent GMP, the IGF-1R phosphorylation declined significantly compared with survivor and non-irradiated control GMP (Figure 2A,B). Similar to GMP, survivor SMP hearts maintained significantly higher levels of IGF-1R phosphorylation compared with the non-irradiated SMP. There was a slight but insignificant reduction in IGF-1R phosphorylation in decedent SMP compared with survivors (Figure 2A,B). Since GMP have significantly higher amounts of heart IGF-1R activation in non-irradiated controls and survivors compared with SMP, these findings show the differential activation status of IGF-1 signaling in the hearts of radiosensitive and radioresistant animals.

The activation of Akt as a downstream target of IGF-1 signaling in the hearts of GMP and SMP presented a pattern moderately similar to that of IGF-1R activation (Figure 2A). In survivor GMP, there was a marginal and insignificant rise in Akt phosphorylation compared to non-irradiated animals. Decedent GMP showed a significant reduction in Akt phosphorylation compared with survivor GMP. SMP survivors showed higher Akt phosphorylation than non-irradiated SMP and there was no significant change in Akt phosphorylation between survivor and decedent SMP (Figure 2A,C).

### 2.2. Plasma NO Level Is Higher in SMP than GMP and Declines in Irradiated Decedent SMP

As discussed above, IGF-1 signaling can regulate the activation of eNOS and production of NO by endothelial cells. Therefore, we next tested whether this variation was associated with differential activation status of eNOS in the heart tissues. We assessed eNOS activation in heart lysates of irradiated survivor or decedent GMP/SMP by immunoblot using an antibody that detects the phosphorylated form of eNOS (p.eNOS Ser1177). The quantification of immunoblots showed no difference in eNOS phosphorylation between irradiated survivors of both strains and between survivors and decedents of the GMP strain, however, for SMP decedents where there was a significant reduction in eNOS phosphorylation compared with SMP survivors (Figure 3A,B). Next, we determined whether there were any inherent variations in plasma NO levels between non-irradiated GMP and SMP and if survival status following H-ARS was associated with changes in the plasma NO levels. Plasma NO measurement in non-irradiated GMP and SMP showed that SMP had significantly higher levels of NO than GMP (Figure 3C). Since NO is a major determinant of cardiovascular health, this finding suggests more effective maintenance of cardiovascular homeostasis in SMP than in GMP. We next evaluated changes in the plasma NO levels of SMP survivors and decedents by comparing their NO levels at euthanasia time points to their respective NO levels measured one day prior to irradiation. In survivors there was no significant decrease in plasma NO levels at necropsy, but NO levels had significantly declined in all decedents (Figure 3D). This finding demonstrated the direct correlation between post-irradiation survival status and higher plasma NO levels.

Since a growing body of evidence has shown that ionizing radiation is capable of inducing oxidative stress [14] and IGF-1 signaling is implicated in counteracting the cytotoxic effects of oxidative stress [11,24,25], we tested whether differential cardiac IGF-1 signaling was associated with changes in the activity of the cardiac antioxidant response genes.

### 2.3. GMP Manifest Weaker Activation of Cardiac Nrf2 Transcription Factor and Higher Levels of Oxidative Stress Marker and Mammalian Target of Rapamycin (mTOR) Activation than SMP

The Keap1-Nrf2 system, comprising the Nrf2 transcription factor and Nrf2-inhibitory protein Keap1, a redox-sensitive E3 ubiquitin ligase and substrate adaptor senses and regulates the cellular response to oxidative stresses. Nrf2 directs the regulation of cellular antioxidant response via inducing the transcriptional activation of a subset of genes involved in adaptation to oxidative stress [26]. To test whether there were any endogenous or radiation-induced changes in the activation of cardiac Nrf2 transcription factor in non-irradiated and irradiated survivor or decedent GMP/SMP, an antibody that recognizes the activated form of Nrf2 was used (p.Nrf2 Ser40). The phosphorylation of Nrf2 on Ser40 by protein kinase C promotes its dissociation from Keap1 and subsequent translocation into nucleus that allows the transcription of antioxidant response genes [27]. Western blot analysis using heart lysates of the control and irradiated samples revealed that non-irradiated GMP animals possessed significant amounts of activated Nrf2 protein which was absent in the SMP (Figure 4A). In all tested irradiated survivor and decedent SMP hearts, Nrf2 was robustly activated, whereas there was no evidence of Nrf2 activation in survivor or decedent GMP hearts compared to hearts obtained from non-irradiated GMP. The lack of Nrf2 activation in GMP survivors and decedent hearts, suggested the absence of effective cardiac Nrf2 activation in GMP strain (Figure 4A,B). Analysis of Keap1, the Nrf2 inhibitor, also revealed a pattern consistent with efficient activation of Nrf2 in SMP. Western blot analysis of heart lysates showed that GMP regardless of irradiation exposure status had higher levels of Keap1 compared with SMP (Figure 4A,C). The higher basal levels of activated Nrf2 in non-irradiated GMP and weaker Nrf2 induction efficiency in irradiated GMP hearts along with higher Keap1 levels compared with SMP suggested that GMP may suffer from higher levels of oxidative stress. 

PA28β is a proteasome activator subunit that is induced in response to oxidative stress by Nrf2 and assists in the degradation of oxidized proteins [28,29]. The western blot analysis of heart lysates showed that compared with SMP, the non-irradiated GMP had significantly higher levels of PA28β expression (Figure 4A,D). This observation was suggestive of higher oxidative stress levels in the GMP hearts compared with SMP.

The serine/threonine kinase, mammalian target of rapamycin (mTOR) mediates energy metabolism and coordinates cell growth in response to PI3K/Akt stimulation and nutritional status of the cells [30]. mTOR activation via phosphorylation on Ser-2448 is subject to regulation by AKT [30] and the p70S6 kinase [31]. mTOR activation alleviates oxidative stress in cardiac tissue and is a crucial regulator of cardiac homeostasis [32,33,34,35]. Therefore, using western blot and an antibody which detects the phosphorylated mTOR (p.mTOR Ser2448), we analyzed the activation status of mTOR in non-irradiated and irradiated minipig heart lysates. It was revealed that non-irradiated GMP had high levels of p.mTOR in their hearts while in non-irradiated SMP there was no evidence of mTOR activation (Figure 4A,E). In SMP, the mTOR phosphorylation was only seen in irradiated survivor and decedent SMP hearts. However, in all GMP the mTOR was phosphorylated regardless of the irradiation status (Figure 4A,E). This finding is suggestive of higher levels of oxidative stress in GMP than in SMP hearts.

Oxidative stress is elicited by elevated intracellular levels of reactive oxygen species (ROS) that may cause damage to lipids, proteins, and DNA. Multiple enzymes such as superoxide dismutase (SOD), glutathione peroxidase (GPx) and catalase assist in ROS detoxification. Catalase is tightly associated with cellular response to oxidative stress and is responsible for catalyzing the breakdown of hydrogen peroxide (H_2_O_2_), which is generated in high levels when cellular levels of ROS increase [36]. Thus, measurement of catalase expression or activity could provide clues about the redox status of any given cell or tissue. To evaluate the catalase activity in heart samples, we measured its expression and enzymatic activity via western blot and catalase activity measurement in heart protein extracts, respectively. Our analyses showed no difference in GMP heart catalase expression or activity between survivors and decedents (Figure 5A–C). In SMP, the decedent animals had slightly higher but insignificant levels of catalase expression and significantly higher activity compared with survivors (Figure 5A–C). Catalase activity comparison between survivor GMP and SMP did not show any significant difference (Figure 5C). The Real-time PCR (or quantitative PCR, q-PCR) analysis of catalase transcripts in GMP and SMP hearts showed a slight but significant reduction in GMP decedents compared with GMP survivors (Figure 5D), with no significant findings in SMP. The heart peroxide measurement also revealed a pattern similar to catalase expression for both minipig strains. There was no difference in heart peroxide levels between survivor and decedent GMP, but in SMP, the decedents manifested significantly higher levels of peroxide than survivors. Moreover, there was no difference in heart peroxide levels between survivors of GMP and SMP (Figure 5E). The heart catalase and peroxide analyses were suggestive of higher levels of oxidative stress in GMP than SMP.

### 2.4. Irradiated Decedent GMP Demonstrate Weaker Induction of HO-1 Gene Expression Compared with SMP

Our analyses so far revealed weaker Nrf2 activation in the GMP hearts than SMP and also provided evidence of higher oxidative stress in GMP hearts than SMP. Thus, it was asked whether GMP could also exhibit differences in the induction potency of Nrf2 target genes that are involved in responding to changes in the redox status of the cell [37]. To address this question, we used q-PCR to assess the expression of Nrf2 target genes, Heme oxygenase-1 (HO-1), NAD(P)H:quinone oxidoreductase 1 (NQO1), Cu/Zn-superoxide dismutase 1 (SOD1), Cytochrome B-245 Alpha Chain (CYBA), Cytochrome B-245 Beta Chain (GP91) in the hearts of irradiated survivor or decedent GMP and SMP. HO-1 is an important component of the antioxidant defense system that catalyzes the oxidative breakdown of heme (molecule with pro-oxidant properties) to Fe^2+^, carbon monoxide (CO), and biliverdin. Biliverdin is subsequently reduced to bilirubin by biliverdin reductase and serves as an antioxidant agent [38]. The q-PCR analysis of HO-1 showed 3.16 ±1.26 (mean ± SEM) upregulation in GMP decedents compared with GMP irradiated survivors, while in SMP, the irradiated decedents manifested 10.5879 ± 7.97 (mean ± SEM) upregulation compared with SMP survivors (Figure 6A). This observation is consistent with enhancement of oxidative stress in decedent hearts and the requirement for HO-1 antioxidant gene response. However, as seen by almost 3-fold difference in HO-1 expression between decedents of the two strains, the SMP are more effective in inducing antioxidant gene response compared with GMP that could suggest better adaptation to oxidative stress in SMP than GMP.

The q-PCR analysis of other Nrf2 target genes, NQO1, SOD1, CYBA, and GP91 in the heart samples from irradiated survivor and decedent GMP were weaker than HO-1 induction in decedents, and the fold changes of expression were insignificant (Figure 6B–E). In SMP hearts, the decedents showed either marginal reduction or no changes in the expression of NQO1, SOD1, CYBA, and GP91 (Figure 6B–E).

### 2.5. Irradiated Survivor or Decedent GMP/SMP Produce Similar Levels of ATP in Their Hearts

Mitochondria as the source of cellular ATP production are heavily susceptible to radiation injury [39], and we showed that IGF-1 signaling in the heart is impacted by radiation. Owing to the importance of IGF-1 signaling in mitochondrial biogenesis and mitophagy [40], we next tested whether ATP availability was affected by irradiation in the heart tissue of the GMP/SMP. Heart ATP measurement did not show any differences between the survivor or decedent animals of both strains (Figure 7), suggesting that in minipig hearts, mitochondrial ATP production or their function is not affected by irradiation.

## 3. Discussion

In the current paper, we show for the first time that dysregulation of cardiac IGF-1 signaling is associated with inter-strain sensitivity to radiation injury in H-ARS model and that post-irradiation survival is accompanied with invigorated cardiac IGF-1 signaling in radiosensitive and radioresistant strains of minipigs. Incapacity to enhance the cardiac IGF-1 activation was associated with death following irradiation exposure in H-ARS. The western analysis showed that non-irradiated control GMP maintained higher levels of heart IGF-1R phosphorylation than SMP. Despite higher levels of IGF-1R activation, the GMP were more sensitive to irradiation than SMP. In the irradiated survivor, GMP, also the heart IGF-1R, phosphorylation was undisputedly higher than the irradiated survivor SMP. This observation provides solid evidence to support that compared with SMP, the GMP heart tissue requires higher rates of IGF-1 signaling to promote survival or maintain cardiac homeostasis not only under stressful challenges such as H-ARS, but also under normal physiological conditions. In this paper, and in our previously published report, we showed that decedent GMP [7] and SMP have higher levels of IGF-1 in their plasma compared with survivors. Therefore, the observed rise in plasma IGF-1 levels in irradiated decedent GMP and SMP, the increased IGF-1R activation in survivor hearts of both strains and the high level of IGF-1R activation in GMP controls compare with SMP controls likely suggests resistance to IGF-1 signaling in GMP strain and in irradiated decedent animals of SMP strain. The interrupted or less efficient cardiac IGF-1R signaling in GMP could either explain or be consistent with their physical dwarfism and their higher susceptibility to metabolic syndrome and H-ARS.

Perhaps the dysregulation of IGF-1 signaling or possible IGF-1 resistance combined with higher levels of background oxidative stress in GMP hearts could be responsible for drastically lower plasma NO levels in GMP compared with SMP. Indeed the majority of NO in circulation originates from eNOS activity in endothelium [41] and its activation via phosphorylation, is regulated by Insulin/IGF-1 signaling [12,42]. The finding that plasma NO levels are only reduced in the irradiated decedent SMP and not in the survivors occurred concomitantly with a slight reduction of IGF-1R activation in the decedent heart tissue. This shares similarly with our previously reported reduction of NO levels in the plasma of the decedent but not survivor GMP [7] where decedents manifest drastic loss of heart IGF-1R activation. Redox status of the cells can impact the levels and bioavailability of essential eNOS cofactor, Tetrahydrobiopterin (BH4) and production of NO by eNOS. BH4 is of central importance in maintaining eNOS in the optimal “coupled” state, which is the form of eNOS involved in NO production. Under pathological conditions, the loss of BH4 bioavailability and resultant eNOS uncoupling may lead to the reduced production of NO and enhanced production of peroxynitrite (ONOO−). Peroxynitrite is a highly reactive molecule that exhibits strong oxidizing properties and may cause the impairment of essential endothelial functions [43]. The measurement of peroxide, catalase expression, and catalase activity in heart lysates of GMP showed no statistically significant difference between irradiated survivors and decedents. However, in SMP, there were significant changes in heart peroxide levels, catalase expression, and catalase activity between irradiated survivors and decedents. This observation suggests that irradiated survivor GMP suffer from higher basal levels of oxidative stress or disturbed redox homeostasis. Furthermore, our subsequent experiments also verified the hypothesis that GMP have a defective antioxidant response system. Our data show comparatively higher levels of the oxidative stress response protein, PA28β, attenuation of Nrf2 activation, increased expression of Keap1, and weaker induction of HO-1 expression in the hearts of GMP than in SMP.

Conceivably the pattern of Nrf2 and mTOR activation, as assessed by phosphorylation of Nrf2 and mTOR in western blot analyses, in non-irradiated and irradiated GMP/SMP is compelling evidence that GMP have higher basal levels of oxidative stress than SMP. This is supported by our western blot analyses showing the similarity in the pattern of mTOR and Nrf2 activation in our experimental model. mTOR and Nrf2 activation were both absent in non-irradiated SMP hearts but present in non-irradiated GMP and in irradiated GMP/SMP. PI3K/Akt/mTOR signaling is necessary for normal regulation of cardiac structure, cardiometabolism, and cardioprotection, reportedly through alleviating the effects of oxidative stress in the heart [33,34,35]. Therefore, the activation of Nrf2 and mTOR in non-irradiated GMP hearts is suggestive of disturbed redox status or higher levels of oxidative stress in GMP. Collectively, our data highlight the association between strain-specific cardiac IGF-1 signaling and radioresistancy in H-ARS model, link susceptibility to H-ARS with dysregulation of cardiac IGF-1 and eNOS signaling, and highlight the role of redox status, the Nrf2 and mTOR activation in response and susceptibility to radiation injury.

## 4. Materials and Methods

### 4.1. Animal Strains, Radiation, Blood, and Tissue Collection from Animals

The procedures related to animal acquisition, housing, irradiation, euthanasia, scoring system for survivors or decedents, and blood—or other bio sample collections—have been described in our previous paper [7]. Briefly, 48 male 3-to-5-month-old Gottingen minipigs weighing 8–12 kg, and 24 Sinclair minipigs with similar age range weighing 10–14 kg, were purchased from Marshall BioResources (North Rose, NY, USA) and Sinclair BioResources (Columbia, MO, USA), respectively. During the experimental procedures, the minipigs were housed in the Veterinary Science Department, Armed Forces Radiobiological Research Institute (AFRRI, Bethesda, MD, USA). In all procedures, we followed the protocols approved by AFRRI’s Institutional Animal Care and Use Committee (IACUC), permit number P-2015-10-008 approved on December 11, 2015 by IACUC chair Dr. David R. Lesser, and the Guide for the Care and Use of Laboratory Animals. The animals were housed individually and maintained on a 12 h light–dark cycle in rooms set at 61–81 Fahrenheit with 30–70% relative humidity. Before irradiation, the animals were anesthetized by single intramuscular injection of a mixture of Telazolt (2 mg/kg) and xylazine (1 mg/kg). The total-body irradiation at doses ranging 1.7–2.3 Gy was achieved by cobalt-60 (^60^Co) gamma photons (~0.6 Gy/min). Real-time dosimetry was performed for every irradiation episode with an ionization chamber located in the radiation field without interfering with radiation beams directed toward the animal. The day of exposure was considered “day 0”. The animals were followed for 45 days post-irradiation while receiving minimum preventive supportive care during the course of the study. Moribund animals were euthanized upon manifesting at least one absolute criteria symptom (non-responsiveness, dyspnea, hypothermia) or a combination of four non-absolute criteria (hypothermia, anorexia, anemia, lethargy, vomiting/diarrhea, vestibular signs and prolonged hemorrhage). Animals that survived the entire study period of 45 days were considered as survivors, and animals euthanized due to radiation-induced health deteriorations were considered decedents.

### 4.2. Western Blot Analysis

Protein from homogenized left ventricles of hearts was extracted using ice-cold Radioimmunoprecipitation assay (RIPA) buffer (ThermoFisher, Rockford, IL, USA) supplemented with protease and phosphatase inhibitors and quantified using Pierce BCA protein assay kit (ThermoFisher, Rockford, IL, USA). Equal amounts of chemically reduced protein lysates were separated using 4–15% gradient polyacrylamide gels, and transferred onto PVDF sheets (BioRad, Hercules, CA, USA). Blocking and incubation with primary and secondary antibodies were carried out using Western Breeze Chemiluminescent kit (ThermoFisher Scientific, Massachusetts, USA). Antibodies: eNOS (sc-376751), catalase (sc-271803) (Santa Cruz Biotechnology, Santa Cruz, CA, USA); AKT (9272), p.AKT (9271), IGF-1R (9750), p.IGF-1R (3024), Keap1 (8047), PA28β (2409), p.mTOR (2971), β-Tubulin (2128), Glyceraldehyde 3-phosphate dehydrogenase (GAPDH) (2118) (Cell Signaling Technology, Danvers, MA, USA); p.eNOS (MAB9028) (R&D Systems, Minneapolis, MN, USA), p.Nrf2 (76026) (Abcam, Cambridge, MA, USA). Densitometric analysis of proteins was performed using Image Lab 5.2.1 Software (BioRad) or ImageJ (NIH, Bethesda, Maryland, USA). β-Tubulin and GAPDH were used as the loading controls for Western blot analyses. For each group of animals the protein lysates from the hearts of at least 7 individual animals were analyzed three times or more by Western blot technique.

### 4.3. ELISA, Catalase Activity, Peroxide, and ATP Measurements

Frozen aliquots of plasma samples were thawed on ice, clarified by centrifugation at 10,000× *g* for 30 min at 4 °C, and were subsequently diluted 1:100 times to measure the concentration of IGF-1 (Biomatik, Wilmington, Delaware, USA).

Catalase activity was assayed in 96-well plates using the colorimetric Amplex™ Red Catalase assay kit (ThermoFisher, Rockford, IL, USA). 1 microgram of each animals’ total heart lysate prepared in RIPA buffer were evaluated according to the provided kit instructions. The plates were read using Spectramax 250 plate reader (Molecular Devices, San Jose, CA, USA) according to recommended settings.

For heart peroxide measurements, 25 mg of frozen heart samples were pulverized on dry ice and homogenized in 300 µL of lysis buffer containing 0.1 M KCl and 0.1 M Na_2_HPO_4_·7H_2_O. Following homogenization using a hand-held homogenizer, the lysates were spun at 14,000× *g* for 15 min at 4 °C and the supernatant was used for peroxide measurement using the Pierce Quantitative Peroxide Assay Kit according to kit instructions.

To measure the heart ATP levels, 20 mg of pulverized heart samples were lysed using Tris-EDTA-saturated phenol (phenol-TE) and hand-held tissue homogenizer. Following spinning at 14,000× *g*, the supernatant was subjected to two rounds of extraction with 400 µL of Chloroform. The supernatant was diluted 1/50 in water and ATP levels were measured using ATP Determination Kit (ThermoFisher).

### 4.4. Nitric Oxide Measurement

For NO measurement, 150 µL of plasma was clarified using Amicon-10kDa spin filters (Millipore, Burlington, MA, USA) by centrifugation at 14,000× *g* for 45 min at 4 °C. 50 µL of the filtrate was assayed in each well of a 96-well microplate according to total Nitric oxide and Nitrate/Nitrite assay kit instructions (R&D Systems). Briefly, first the nitrate in samples is enzymatically converted to nitrite by nitrate reductase. In the second step, the concentration of the nitrite is determined by colorimetric Griess Reaction that is calculated by plotting unknown sample values against a standard curve for nitrite samples (1–10 µM range).

### 4.5. Real-Time PCR

For quantitative PCR analysis, 25 mg of each frozen heart sample was pulverized and lysed in 800 µL of QIAzol (Qiagen) lysis reagent. The mix was vigorously vortexed at maximum power for 30 s. Subsequently, the supernatant was collected after two rounds of chloroform extraction and spinning. The supernatant was used for RNA extraction following the instructions of the RNeasy plus universal mini kit columns (Qiagen). The quantitative and qualitative assessment of the RNA samples was performed using NanoDrop C (ThermoFisher) and 1 µg of total RNA was used for cDNA synthesis. The cDNA was synthesized using SuperScript™ IV First-Strand Synthesis System (ThermoFisher) following the recommended steps in a standard thermocycler machine (ThermoFisher). For Real-time PCR, 10 µL of TaqMan™ Fast Advanced Master Mix, 1 µL of TaqMan probe (both from ThermoFisher), 1 µL of synthesized cDNA and 8 µL H_2_O were mixed in thin-walled, RNase-free 100 µL PCR plates and assayed in QuantStudio3 Real-time PCR machine (ThermoFisher). For each sample, the PCR was run in triplicate wells and GAPDH gene was used as reference for quantification. The acquisition and analysis of Real-time PCR data was achieved by QuantStudio3 Design and Analysis Software V1.4.3 from Applied Biosystems. To calculate the changes in the transcript numbers for any target gene, the cycle threshold (Ct) values for GAPDH gene were subtracted from tested genes and fold changes were calculated by comparison of Ct values in decedent animals to survivor animals following the vendor instructions (ThermoFisher). The fold change of decedent samples that were greater than ±2.5 fold of survivor samples were considered significant.

### 4.6. Statistical Analysis

Data analysis was performed using either GraphPad Prism or Microsoft Office 2013 Excel. Data are presented as Mean ± SEM of all tested groups. The Student’s t-test for independent samples was used to determine the significance of the differences between tested groups.

## Figures and Tables

**Figure 1 ijms-21-05049-f001:**
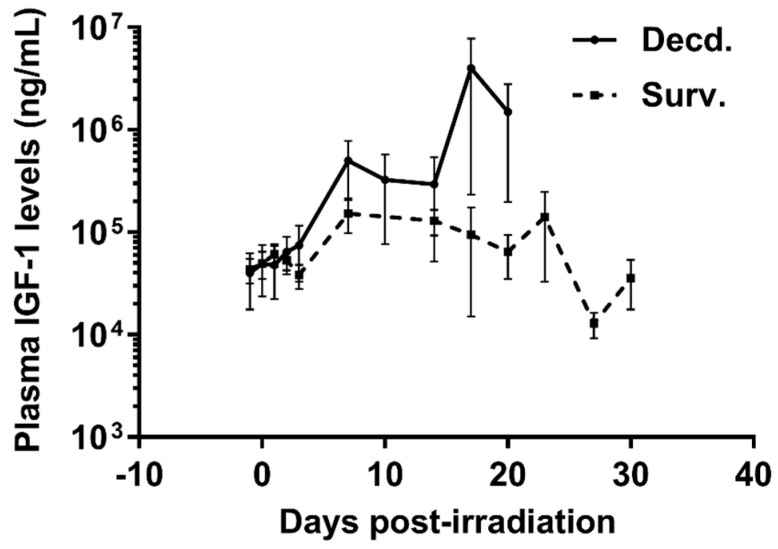
Plasma Insulin-like Growth Factor-1 (IGF-1) levels are increased in irradiated decedent Sinclair minipig (SMP). Plasma IGF-1 levels were assayed by ELISA assay at indicated days following irradiation until necropsy in survivor (Surv. *n* = 9) and decedent (Decd. *n* = 4) SMP. Note that in the days 17 and 20 post-irradiation there are measurements available from only two decedent animals.

**Figure 2 ijms-21-05049-f002:**
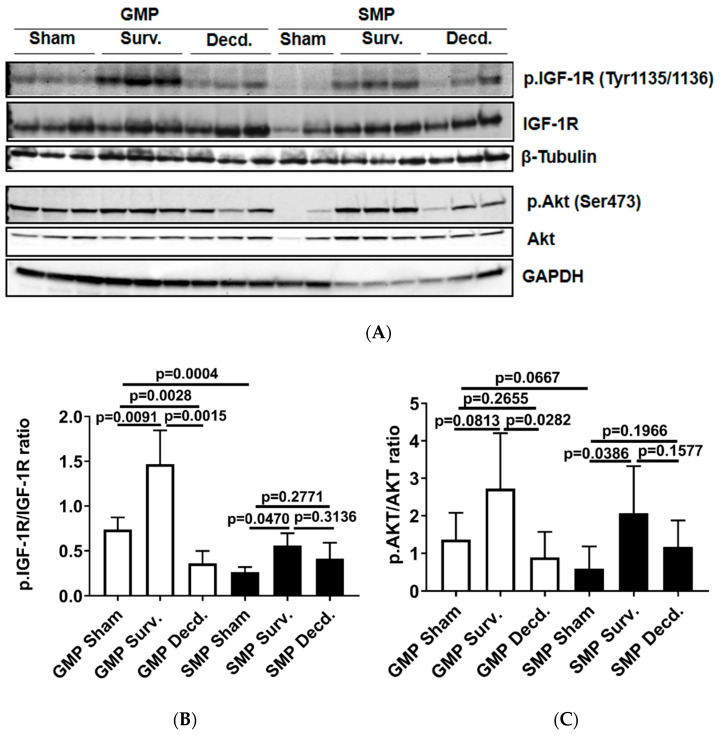
Cardiac IGF-1 receptor (IGF-1R) is differentially regulated in Gottingen minipig (GMP)/SMP strains and is significantly impacted by irradiation. (**A**) Western blot analysis of phosphorylated IGF-1R and Akt in heart protein lysates of non-irradiated control and irradiated survivor (Surv.) or decedent (Decd.) GMP and SMP. Glyceraldehyde 3-phosphate dehydrogenase (GAPDH) and β-Tubulin blots are included as loading controls. (**B**,**C**) Graphs representing the analysis of IGF-1R and Akt phosphorylation in the protein lysates of control and irradiated animals by western blot.

**Figure 3 ijms-21-05049-f003:**
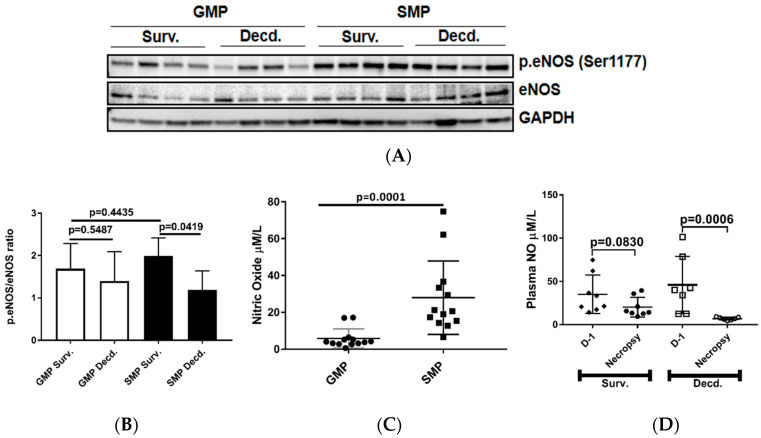
Analysis of cardiac endothelial nitric oxide synthase (eNOS) activation by phosphorylation and plasma NO level in unirradiated and irradiated SMP and GMP. (**A**) Western blot analysis showing the phosphorylation status of eNOS (Ser1177) in heart protein extracts of irradiated survivor and decedent SMP and GMP. GAPDH is shown as loading control. (**B**) Graph representing the quantification of eNOS phosphorylation (Ser1177) in the protein lysates of irradiated survivor and decedent animals by western blot. (**C**) Comparison of plasma NO level in non-irradiated male SMP and GMP. Each circle or square represents the measured values from one individual animal. (**D**) Comparison of plasma NO levels in survivor and decedent SMP one day before irradiation (Day-1, D-1) and at scheduled (day 45 for survivors) or unscheduled necropsy days for decedents. Each circle or square represents the measured values from one individual animal.

**Figure 4 ijms-21-05049-f004:**
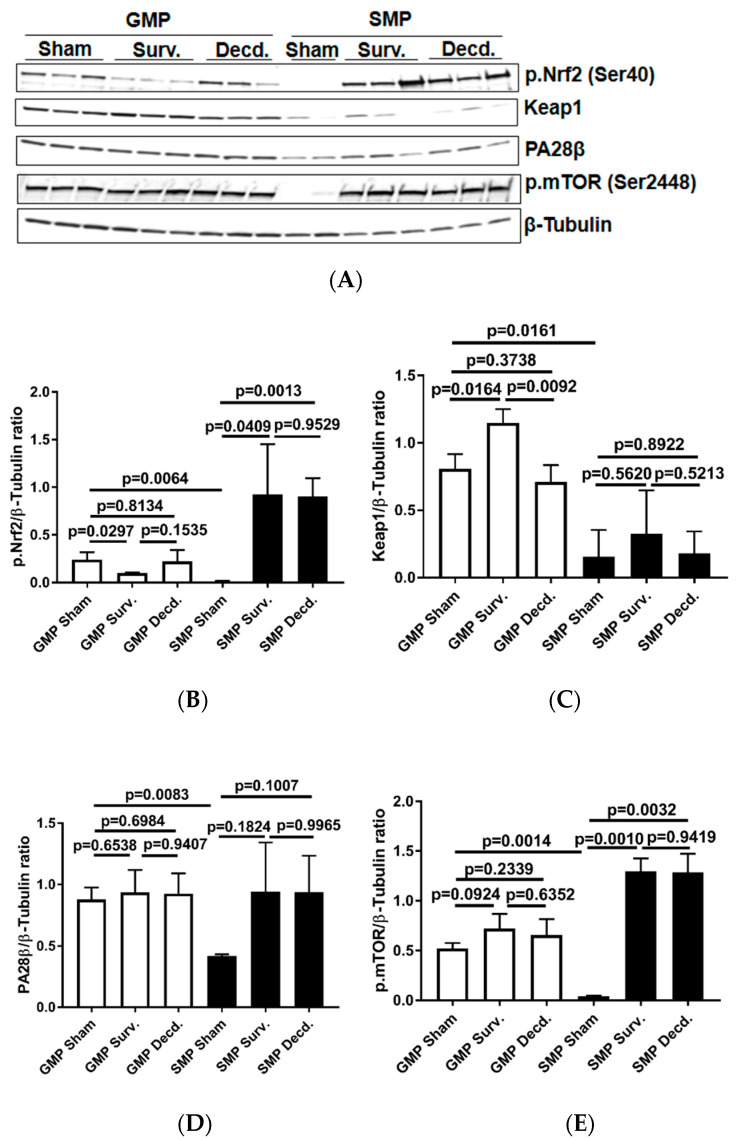
Analysis of nuclear factor erythroid 2–related factor 2 (Nrf2) activation, Keap1, mammalian target of rapamycin (mTOR), and PA28β in the hearts of sham and irradiated GMP/SMP. (**A**) Western blot analysis of activated Nrf2 (Ser40) transcription factor, Keap1, PA28β, and phosphorylated mTOR (p.mTOR) in heart lysates of sham and irradiated survivor or decedent GMP/SMP. β-Tubulin is shown as loading control. (**B**–**E**) Graphs representing the quantification of Western blot analysis of Nrf2 phosphorylation, Keap1, PA28β, and mTOR phosphorylation in the protein extracts of control and irradiated GMP/SMP animals. Y-axis in each graph shows the normalized ratio of respective Western blot bands to β-Tubulin as loading control.

**Figure 5 ijms-21-05049-f005:**
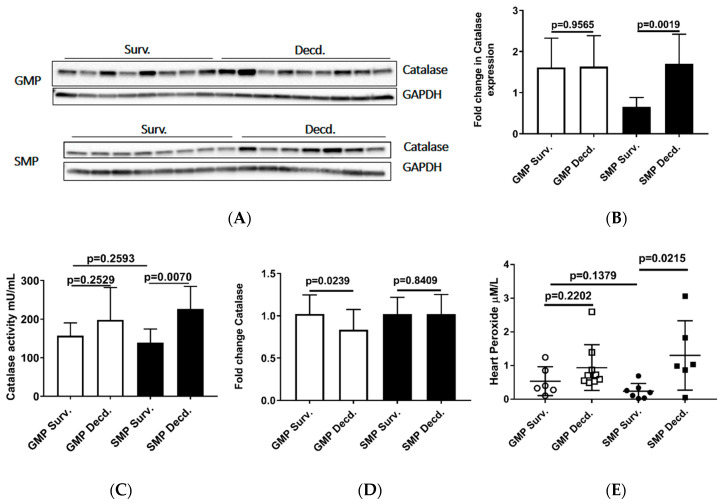
Analysis of Catalase and Peroxide in the heart protein extracts from irradiated survivor and decedent GMP/SMP. (**A**) Western blot analysis of Catalase expression in heart lysates of irradiated survivor or decedent SMP/GMP. β-Tubulin is shown as loading control. (**B**) Statistical analysis of Western blot bands intensity following normalization to GAPDH as loading control. (**C**) Measurement of Catalase enzymatic activity in heart protein extracts from irradiated survivor or decedent GMP/SMP. (**D**) Quantitative PCR analysis of catalase transcripts in hearts from irradiated survivor or decedent GMP/SMP. (**E**) Graph representing the Peroxide levels measured in heart lysates of irradiated survivor or decedent SMP/GMP. Each solid circle or square represents measurement from one individual animal.

**Figure 6 ijms-21-05049-f006:**
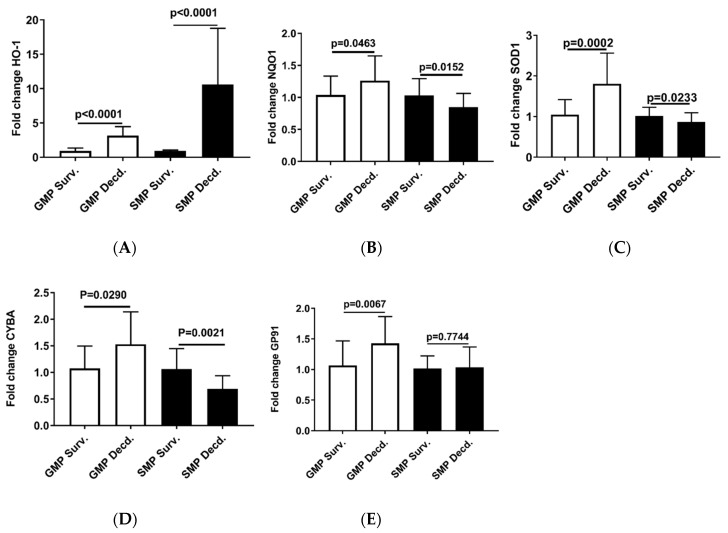
Real-time PCR analysis of Nrf2 target genes in heart cDNA prepared from irradiated survivor and decedent GMP/SMP. (**A–E**) Normalized quantitative PCR analysis showing the fold change in the expression of HO-1, NQO1, SOD1, CYBA, and GP91 in the hearts of irradiated survivor or decedent GMP/SMP. The y-axis represents the fold changes in expression of decedent animals genes compared with the survivors.

**Figure 7 ijms-21-05049-f007:**
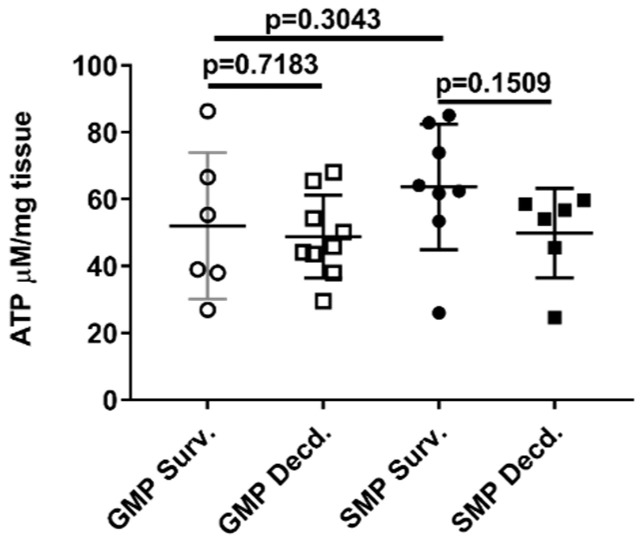
Heart ATP levels in irradiated survivor and decedent GMP/SMP. Graph representing the ATP levels measured in heart lysates of irradiated survivor or decedent SMP/GMP. Each circle or square in the graph represents measurements from one individual animal.

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
