# Peer review of "Dysregulated Cardiac IGF-1 Signaling and Antioxidant Response Are Associated with Radiation Sensitivity"

_ijms, 2020, doi:10.3390/ijms21145049_

Round 1

Reviewer 1 Report

The present paper titled "Dysregulated cardiac IGF-1 signaling and antioxidant response are associated with radiation sensitivity" is result of descriptive study without control experiments. The design of experiments is very confusing and results of these experiments are questionable. Authors need to perform a significant work to make this manuscript publishable.

Major flaws:

First of all, authors improperly used term H-ARS model.  H-ARS is characterized by severe neutropenia and thrombocytopenia, and possible death due to infection and/or bleeding. In introduction authors stated "In our previous studies, complete blood count (CBC) profiles of H-ARS survivor and decedent animals from both GMP and SMP strains were similar. Thus, disorders caused by hematopoietic effects of radiation exposure do not appear to be the cause of differential radiosensitivity in GMP and SMP" (lines 69-71). This can not be applied to H-ARS model.

If authors hypothesize that sensitivity to total body irradiation caused by cardiac IGF-1 signaling, they have to conduct control experiments IGF-1R inhibitors, IGF-1 mAb, or any other approaches to modify cardiac IGF-1 signaling pathway.

Figure 1 demonstrate significantly higher level of plasma IGF-1 in irradiated decedent SMP comparing with non-irradiated SMP. Survivors of SMP strain show no difference in the level of plasma IGF-1 comparing with non-irradiated SMP. At the same time, figure 2 shows completely opposite effect for cardiac IGF-1R activation. It looks like the lower level of plasma IGF-1 corresponds to the higher cardiac IGF-1R activation. Authors dis not unravel these confused data.

Graphs representing Western bolt analysis on Figures 2, 3, 4 demonstrate p-values. However, statistical analysis can not be applied to very small groups which contain only 2-4 samples. Hence, all these p-values are unreliable.

Plasma concentration of NO is not correlated with NO concentration in any tissues. Hence, plasma level of NO shown on Figure 3C, 3D can not be extrapolated to eNOS activation or NO level in cardiac tissue.

QPCR analysis on Figures 6B, 6C, 6D, 6E shows very low p-values. However, for qPCR analysis expression difference less than 2-fold is never accepted as significant.

Author Response

First of all, authors improperly used term H-ARS model.  H-ARS is characterized by severe neutropenia and thrombocytopenia, and possible death due to infection and/or bleeding. In introduction authors stated "In our previous studies, complete blood count (CBC) profiles of H-ARS survivor and decedent animals from both GMP and SMP strains were similar. Thus, disorders caused by hematopoietic effects of radiation exposure do not appear to be the cause of differential radiosensitivity in GMP and SMP" (lines 69-71). This can not be applied to H-ARS model.

Response: We believe that in the current manuscript the term H-ARS indeed applies to our study since in previously published papers by our team and others, for instance in the papers authored by Moroni et al., 2013, PMID: 23845847, Kaur et. al., DOI: 10.21175/RadJ.2017.02.017, using same doses of radiation it was shown that the doses used in this study cause hematopoietic progenitor depletion, thus the cause of the death for the decedent animals is a combination of hematopoietic failure (thrombocytopenia, neutropenia) and dysregulated IGF-1 signaling in heart and other tissues.

Page 78 of the second manuscript (Kaur et al.,) in second paragraph “On average, all animals (survivors and decedents) developed neutropenia (ANC ˂500/µl) and thrombocytopenia (platelets <20,000/µl) around day 14 post-irradiation. Prevalence and duration of neutropenia and thrombocytopenia were dose- dependent, and lasted for 2 to 4 weeks. Full recovery was not reached within the 45-day observation period.”

We have modified the aforementioned sentence "In our previous studies, complete blood count (CBC) ……. similar. As follows: “In our previous studies, kinetics of complete blood count (CBC) profiles ……………were similar (7, 15, unpublished data).

Page 78 of the above manuscript in second paragraph “On average, all animals (survivors and decedents) developed neutropenia (ANC ˂500/µl) and thrombocytopenia (platelets <20,000/µl) around day 14 post-irradiation. Prevalence and duration of neutropenia and thrombocytopenia were dose- dependent, and lasted for 2 to 4 weeks. Full recovery was not reached within the 45-day observation period.”

If authors hypothesize that sensitivity to total body irradiation caused by cardiac IGF-1 signaling, they have to conduct control experiments IGF-1R inhibitors, IGF-1 mAb, or any other approaches to modify cardiac IGF-1 signaling pathway.

Response: In the current manuscript we do not mention hypothesizing or concluding that IGF-1 signaling dysregulation is the cause for moribundity of the irradiated animals. We rather report an association between moribundity following TBI and cardiac IGF-1 signaling dysregulation. In this study since we have seen an increase in serum IGF-1 ligand in the decedent GMP in our previous study (Ref. No. 7) and SMP (current study), it suggests that probably animals have IGF-1 signaling resistance in heart or possibly other organs. We assume that the approach of external interference of IGF1 signaling with inhibitors or mAB to modify the systemic IGF-1 signaling would introduce an error or complexity to this study by impairing the IGF-1 signaling in other organs. Moreover, these experiments are not feasible with large animals in our set up. The precise analysis of the role of cardiac IGF-1 signaling in radiation sensitivity requires targeted deletion of IGF-1R in heart using a mouse model would feasible and it is our future research goal.

Figure 1 demonstrate significantly higher level of plasma IGF-1 in irradiated decedent SMP comparing with non-irradiated SMP. Survivors of SMP strain show no difference in the level of plasma IGF-1 comparing with non-irradiated SMP. At the same time, figure 2 shows completely opposite effect for cardiac IGF1R activation. It looks like the lower level of plasma IGF-1 corresponds to the higher cardiac IGF-1R activation. Authors did not unravel these confused data.

Response: This observation is indeed truly authentic and can be explained in the light of resistance towards specific signaling molecules where there is abundant amount of the ligand but the receptor is unable either to respond and/or propagate the signal to downstream mediators (similar to insulin-resistant subjects).

In the discussion we have mentioned this “Therefore, the observed rise in plasma IGF-1 levels in irradiated decedent GMP and SMP, the increased IGF-1R activation in survivor hearts of both strains and the high level of IGF-1R activation in GMP controls compare with SMP controls, likely suggests resistance to IGF-1 signaling in GMP strain and in irradiated decedent animals of SMP strain. The interrupted or less efficient cardiac IGF-1R signaling in GMP could either explain or be consistent with their physical dwarfism, and their higher susceptibility to metabolic syndrome and H-ARS.”

Graphs representing Western bolt analysis on Figures 2, 3, 4 demonstrate p-values. However, statistical analysis can not be applied to very small groups which contain only 2-4 samples. Hence, all these p-values are unreliable.

In figures 2,3 and 4 the p-values represent combined values from at least 7 animals which we have described in the materials and methods section of the manuscript“For each group of animals the protein lysates from the hearts of at least 7 individual animals were analyzed three times or more by Western blot technique.”

Plasma concentration of NO is not correlated with NO concentration in any tissues. Hence, plasma level of NO shown on Figure 3C, 3D can not be extrapolated to eNOS activation or NO level in cardiac tissue.

In the current manuscript we only present the NO concentrations from plasma samples and no other tissues are presented. It is not clear what the reviewer refers to by “NO concentrations in any tissue”. However, we were aware that measurement of NO levels in heart tissue would provide better clues about cardiovascular homeostasis in the minipigs and measured the heart NO levels using the same kit which we used for plasma NO measurement. However, using different concentrations of heart tissue extracts, we could only measure barely detectable levels of NO and the values were minuscule (not shown in this manuscript). This is consistent with the nature of the NO as a gaseous mediator which is abundant only in the blood where it travels to target tissues or cells.

As we have discussed in the 2nd paragraph of the discussion, the bioavailability of NO is affected by different factors; “Redox status of the cells can impact the levels and bioavailability of essential eNOS cofactor, Tetrahydrobiopterin (BH4) and production of NO by eNOS. BH4 is of central importance in maintaining eNOS in the optimal “coupled” state which is the form of eNOS involved in NO production”. The activation status of eNOS is one of the many factors involved in NO production and its activation status may not necessarily correlate with NO production levels.

QPCR analysis on Figures 6B, 6C, 6D, 6E shows very low pvalues. However, for qPCR analysis expression difference less than 2-fold is never accepted as significant.

As we have mentioned in the manuscript the most significant change in the expression of the Nrf2 target genes was that of HO-1. We are aware that fold changes less than 2.5-fold are not accepted as significant and we do not focus profoundly on the expression of NQO1, SOD1, CYBA and GP91 but rather on H0-1. To highlight and clarify this important, we have modified the material and method section related to qPCR analysis by adding the sentence on the statistical analysis and normalization of ct values.

Reviewer 2 Report

The aim of this study is to investigate “the association between the cardiac IGF-1 signaling pathway and radiosensitivity ”. The Gottingen minipig and Sinclair minipig were irradiated using gamma photons 1.7-2.3 Gy doses). Authors concluded that “that inter-strain radiosensitivity and survival outcomes in H-ARS were associated with activation status of the cardiac IGF-1 signaling and Nrf2 – mediated induction of antioxidant gene expression”. It is very interesting paper. I have significant specific comments listed below:

It has been not clearly presented why the authors choose the cardiac tissue samples (in fact left ventricles) to investigate the effect of irradiation on IGF-1 signaling and antioxidant status.

In my opinion, it should be stated and discussed the aetiology of some inevitable complications of radiation therapy. The radiation-induced heart disease is one of the most serious complications.  It is therefore important to measure contractile function of the heart and factors in confirming the induction of radiation-induced cell injury.

Authors need to explain the novel aspects of this study as H-ARS impact on systemic IGF-1 and the cardiac IGF-1 signaling in fatally irradiated animals has been previously shown.

A limitation of the present study is the lack of the myocardial structure measurements (Heart mass or HM/BM ratio) in relation to radiosensitivity and the cardiac IGF-1 signaling. Authors should comment about this point.

The authors need to explain how IGF-1 binding proteins attitudes to IGF-1 bioavailability and mediates cardiovascular performance. This question has not been explored and described.

Author Response

It has been not clearly presented why the authors choose the cardiac tissue samples (in fact left ventricles) to investigate the effect of irradiation on IGF-1 signaling and antioxidant status.

Response: In the abstract we mention “Since Insulin-like Growth Factor-1 (IGF-1) signaling is associated with the regulation of cardiovascular system homeostasis, we investigated the link between dysregulation of cardiac IGF-1 signaling and radiosensitivity”. IGF-1 is a known radioprotector and is an important regulator of cardiovascular system homeostasis. Since two different minipig strains in our study have different body growth indexes, which itself is potentially regulated by GH/IGF-1 axis, we tested if radiosensitivity was linked to the quality of cardiac IGF-1 signaling. Radiation induced bleeding indicates of cardiovascular damage. We selected heart for study is because we were searching for the likely causes for TBI-related death other than hematopoietic effects of radiation, and we already know that heart is affected by irradiation (refs. 7, 10), therefore we investigated the differences between the two strains of minipigs in the cardiovascular homeostasis which itself is regulated by IGF-1 signaling would explain radiosensitivity in two strains of minipigs.

The left ventricle pumps the red oxygenated blood via the aortic valve into the aorta and supplies oxygen-rich blood to the entire body. Heart undergoes ventricular chamber remodeling in response to pathological conditions such as increased blood pressure (pressure overload). This results in increased ventricular (left) wall thickness due to cellular and interstitial changes that manifest in clinical changes such as increased ventricular thickness cardiac/hypertrophy and heart failure.  Thus measuring the activity of the IGF-1 signaling in left ventricle could potentially reflect the overall wellbeing of the cardiac tissue.

The analysis of the oxidative stress was chosen because we found that plasma NO levels in non-irradiated GMP were lower than SMP. Since NO production can be vastly affected by redox status of the organ, we investigated if differences in oxidative stress were parallel with NO levels, the quality of IGF-1 signaling and differential radiation sensitivity.

In my opinion, it should be stated and discussed the aetiology of some inevitable complications of radiation therapy. The radiation-induced heart disease is one of the most serious complications.  It is therefore important to measure contractile function of the heart and factors in confirming the induction of radiation-induced cell injury.

Response: We definitely agree with the reviewer and will perform this experiment in future if the funding is available but our current study is simply a molecular-biology based mechanistic study which addresses the molecular level changes associated with inter-strain radiation sensitivity. We are interested to perform an in depth analysis of the cardiac contractility function and to address a part of this question, we measured ATP levels in our samples (Figure 7). Completing the supplementary study depends on receiving funding and performing the electrophysiological/electro cardiology analysis of the heart function in collaboration with experts in cardiovascular biology.

Authors need to explain the novel aspects of this study as HARS impact on systemic IGF-1 and the cardiac IGF-1 signaling in fatally irradiated animals has been previously shown.

Response: We believe that in the introduction, results and discussion section of the manuscript we have addressed issues related to this comment. Firstly, the study was performed to understand how cardiac IGF-1 signaling is affected in radiosensitive and radioresistant strains, secondly we show that both strains manifest different levels of NO in their plasma and oxidative stress in their heart and lastly we show that efficiency of the antioxidant response mediated by Nrf2 transcription factor is higher in radioresistant strain compared with the radiosensitive strain.

A limitation of the present study is the lack of the myocardial structure measurements (Heart mass or HM/BM ratio) in relation to radiosensitivity and the cardiac IGF-1 signaling. Authors should comment about this point.

Response: We have addressed this comment in the above reviewer comment and in future we will study this aspect if the funds are available. Our study is a molecular levels analysis trying to understand the mechanisms of inter-strain radiation sensitivity.

The authors need to explain how IGF-1 binding proteins attitudes to IGF-1 bioavailability and mediates cardiovascular performance. This question has not been explored and described.

Response: The authors are aware that many factors (including IGF-1 binding proteins, IGFBPs) are involved in the ligand-to-receptor binding efficiency and activation quality of the RTK signaling pathways. However, in the current study the most important and central question was the intrinsic properties of IGF-1 signaling in the cardiac tissue and we have evaluated this quality essentially by measuring the IGF-1R phosphorylation and the downstream Akt activation in heart lysates. To evaluate systemic changes in IGF-1 signaling we have also measured the IGF-1 ligand in the plasma. Measuring the levels of different IGFBPs would provide number of different data set and complicate the interpretation of tissue-specific IGF1 signaling analysis and is not vastly relevant to cardiac-specific IGF-1 signaling.

Round 2

Reviewer 1 Report

The present paper can be accepted in the present form.